# The Synthesis of Manganese Hydroxide Nanowire Arrays for a High-Performance Zinc-Ion Battery

**DOI:** 10.3390/nano12152514

**Published:** 2022-07-22

**Authors:** Jiangfeng Gong, Bingxin Zhu, Zhupeng Zhang, Yuanyuan Xiang, Chunmei Tang, Qingping Ding, Xiang Wu

**Affiliations:** 1College of Science, Department of Physics, Hohai University, Nanjing 210098, China; zhubingxin20211226@163.com (B.Z.); zzp18752006001@163.com (Z.Z.); yyxiang@hhu.edu.cn (Y.X.); 2Ames Laboratory, Department of Physics and Astronomy, Iowa State University, Ames, IA 50011, USA; qingping.ding@gmail.com; 3School of Materials Science and Engineering, Shenyang University of Technology, Shenyang 110870, China

**Keywords:** manganese hydroxide, aqueous zinc-ion battery, surfactant assistant, electrodeposition

## Abstract

The morphology, microstructure as well as the orientation of cathodic materials are the key issues when preparing high-performance aqueous zinc-ion batteries (ZIBs). In this paper, binder-free electrode Mn(OH)_2_ nanowire arrays were facilely synthesized via electrodeposition. The nanowires were aligned vertically on a carbon cloth. The as-prepared Mn(OH)_2_ nanowire arrays were used as cathode to fabricate rechargeable ZIBs. The vertically aligned configuration is beneficial to electron transport and the free space between the nanowires can provide more ion-diffusion pathways. As a result, Mn(OH)_2_ nanowire arrays yield a high specific capacitance of 146.3 Ma h g^−1^ at a current density of 0.5 A g^−1^. They also demonstrates ultra-high diffusion coefficients of 4.5 × 10^−8^~1.0 × 10^−9^ cm^2^ s^−1^ during charging and 1.0 × 10^−9^~2.7 × 10^−11^ cm^−2^ s^−1^ during discharging processes, which are one or two orders of magnitude higher than what is reported in the studies. Furthermore, the rechargeable Zn//Mn(OH)_2_ battery presents a good capacity retention of 61.1% of the initial value after 400 cycles. This study opens a new avenue to boost the electrochemical kinetics for high-performance aqueous ZIBs.

## 1. Introduction

With the growing environmental concerns, the development of rechargeable batteries with high electrochemical energy storage technology is highly needed to meet the requirements of renewable and sustainable energy storage, such as wind or solar power. Lithium-ion batteries (LIBs) are considered as the most promising power sources for their superior cycle performance and high-energy density [1,2]. Although LIBs have been commercialized, some crucial problems, such as high price, low power density and safety concerns, still exist [3,4]. Aqueous rechargeable batteries focused on multivalent metal particles were extensively studied to resolve these drawbacks due to their high safety (aqueous electrolytes) and energy (multi-purpose transporters) efficiencies. The renaissance of aqueous zinc-ion batteries (ZIBs) shows encouraging choices for next-generation energy storage devices for their excellent safety and environmental aspects [5,6,7,8], along with other advantages of low cost, low redox potential and high capacity of the Zn anode.

It is a system engineered to promote the development of ZIBs with great commercial prospects. To improve the intrinsic safety of ZIBs, some scientists are devoted to suppressing the growth of zinc dendrites [9,10,11]. Current research efforts are mainly focused on exploring high-performance cathode materials of ZIBs, such as manganese-based [12,13,14,15,16,17,18,19] and vanadium-based materials [20,21], and so on [22,23,24,25]. Manganese-based oxides are the most favorable cathode materials for aqueous ZIBs because of these advantages. First, a manganese ion consists of various oxidations states (Mn^2+^, Mn^3+^, Mn^4+^); MnO_2_ can accommodate one Zn^2+^ insertion per formula with a high theoretical capacity of approximately 616 mA h/g, in which the Mn^4+^ is reduced to Mn^2+^ [14]. Second, MnO_2_ exists in various crystal forms. In these structures, the basic structural MnO_6_ octahedra units are connected to each other by a co-angle/co-edge, constructing chain, tunnel and layered structures with enough space accommodating foreign cations. The advantages of these structures are the high availability of a Zn^2+^ storage site, increased electrolyte penetration and more favorable surface/interfacial properties [26]. In 2012, Kang’s group reported a zinc-ion storage strategy by utilizing α-MnO_2_ as an effective cathodic electrode for the first time [27]; since then, manganese-based materials have been widely studied and demonstrated as a compelling candidate material for aqueous ZIBs. For example, Wu et al. reported an aqueous Zn/MnO_2_ battery using α-MnO_2_/graphene scrolls as the cathode material, and the cathode delivered a prominent capacity of 362.2 mA h g^−1^ at a current density of 0.3 A g^−1^ with long-term cyclability [10]. Some other works also show similar results [13,15,26,28]. In such electrochemical systems, the typical chemical conversion reaction mechanism of the cathodic MnO_2_ is one electron redox (Mn^4+^ to Mn^3+^) in the conventional natural aqueous electrolyte: MnO_2_ + H^+^ + e^−^ ↔ MnOOH [29]. Inspired by this equation, Zhang et al. synthesized γ-MnOOH nanorods that possess an enhanced electrochemical performance with a specific capacity of 965 mA h g^−1^ at a current density of 200 mA g^−1^ for lithium-ion batteries [30]. As a layered material, manganese hydroxide possesses a large interlayer spacing, which provides a rapid transport pathway for efficient Zn^2+^ insertion/extraction. As a matter of fact, there is a vital consideration for choosing suitable manganese hydroxide or transition metal hydroxide hosts as the cathode in ZIBs [31,32,33,34,35,36,37]. Binders and conductive carbon are usually used as additives to prepare conventional powder electrodes. The additives may generate “dead volume” due to the bonding process of the active material to the substrate. It is necessary to develop a feasible way to manufacture binder-free cathodes to improve the electrochemical performance of ZIBs.

In this study, binder-free electrode Mn(OH)_2_ nanowire arrays are facilely synthesized via electrodeposition. The nanowires were vertically aligned on a carbon cloth. The configuration is beneficial to electron transport during the redox reaction. Meanwhile, the free space between the nanowires may provide more ion diffusion pathways. The as-prepared Mn(OH)_2_ nanowire arrays were used as cathodes to fabricate a high-performance rechargeable aqueous ZIBs with 2 M Zn (CF_3_SO_3_)_2_ serving as the electrolyte. As a result, Mn(OH)_2_ nanowire arrays yielded a high specific capacitance of 146.3 mAhg^−1^ at a current density of 0.5 A g^−1^. It also demonstrates ultra-high diffusion coefficients of 4.5 × 10^−8^~1.0 × 10^−9^ cm^2^ s^−1^ during charging and 1.0 × 10^−9^~2.7 × 10^−11^ cm^−2^ s^−1^ during discharging processes, which are one or two orders of magnitude higher than in other reported studies. Furthermore, the rechargeable Zn//Mn(OH)_2_ battery presents a good capacity retention of 61.1% of the initial value after 400 cycles. Our work presents an inspiring solution to develop a binder-free electrode with a simple surfactant-assistant electroplate method. This work opens a new avenue to boost the electrochemical kinetics for high-performance aqueous ZIBs

## 2. Materials and Methods

### 2.1. Materials

All raw materials purchased were of analytical grade and were not purified before use. Manganese nitrate Mn(NO_3_)_2_·4H_2_O (CAS No.10377-66-9) was purchased from Nanjing Reagent Co. (Nanjing, China); ethylene glycol (EG, C_2_H_6_O_2_, CAS No. 107-21-1), sodium dodecyl sulfate (SDS, C_12_H_25_NaO_4_S, CAS No. 151-21-3) and absolute ethanol (C_2_H_5_OH, CAS No. 64-17-5) were purchased from Kelong Chemical Reagent Co. (Chengdu, China); and the carbon cloth (CC, HCP330N) and glass-fiber filter paper (Whatman GF/F) were purchased from Guangdong Canrd New Energy Technology Co. (Dongguan, China). All electrolytes were prepared with distilled water.

### 2.2. Preparation of SDS-Mn(OH)_2_ on CC

The electrodeposition of Mn(OH)_2_ nanowire arrays was performed under potentiostatic mode with a thermostatic control system. A conventional three-electrode system was used to deposit Mn(OH)_2_ nanowire films assisted with SDS confinement. A platinum electrode and standard saturated calomel electrode (SCE) (saturated KCl) were used as counter and reference electrodes, respectively. The carbon cloth (CC) was cut into a circular shape with a diameter of 1.4 cm. After being washed and dried with ethanol and de-ionized with water several times, it was used as the working electrode. The plating solution was composed of 5% SDS, 20% EG and 0.05 M Mn(NO_3_)_2_. The electrochemical deposition was conducted at a voltage of −1.3 V with a temperature of 70 °C for 30 min. Finally, the deposited samples were immersed in ethanol and de-ionized water, in turn, and dried in an oven at 60 °C for 2 h. The mass loading of manganese hydroxide was 2.4 mg cm^−1^.

### 2.3. Structural Characterization

The crystal structure of the Mn(OH)_2_ nanowire array was analyzed by X-ray diffractometer (XRD, D8 ADVANCE, Bruker, Karlsruhe, Germany). The morphologies and microstructures of the Mn(OH)_2_ nanowires were examined by scanning electron microscope (SEM, Gemini 500, Zeiss, Oberkohen, Baden-Wurttemberg, Germany) and transmission electron microscope (TEM, Tecnai F20, FEI, Hillsborough, OR, USA). The composition of Mn(OH)_2_ was characterized by X-ray photoelectron spectroscopy (XPS ESCALAB 250 Xi, Thermo Fisher Scientific, Waltham, MA, USA).

### 2.4. Fabrication of CC@Mn(OH)_2_//Zn Coin Cells

Electrochemical measurements were performed using CR2032 coin-type cells. The cells were assembled using the CC@Mn(OH)_2_ composite as cathode, zinc metal foil as anode and glass fiber as the separator. A 2 M Zn(CF_3_SO_3_)_2_ aqueous electrolyte was used as the electrolyte. An electric-pressure battery-sealing machine (MSK-160E) was used to seal the cell; the sealing pressure was kept at 0.600 t. Finally, the encapsulated cells were shelved for 12 h.

### 2.5. Electrochemical Tests

All tests were measured using the assembled 2032-coin cell. Cyclic voltammetry (CV), electrochemical impedance spectroscopy (EIS) and galvanostatic charge–discharge (GCD) were performed using the electrochemical workstation (CHI660, Chenhua, Shanghai, China). CV was conducted at virous scan rates under a voltage window of 0.3–1.8 V. EIS was performed at open-circuit potential in a frequency range from 0.01 Hz to 10 kHz with a perturbation of 5 mV. The rate performance, cycle life and galvanostatic intermittent titration technique (GITT) were studied by using the Neware testing system (CT-4000, Neware, Shenzhen, China).

## 3. Results and Discussion

The Mn(OH)_2_ nanowire arrays were prepared on CC through a facile electrodeposition method, in which SDS was used as a templating agent. The electrodeposition of manganese hydroxide proceeded through the following reactions [38]:NO_3_^−^ + 7H_2_O + 8e^−^ → NH_4_^+^ + 10OH^−^(1)
Mn^2+^ + 2OH^−^ → Mn(OH)_2_(2)

The addition of the surfactant during electrodeposition played a vital role in modifying the structure of Mn(OH)_2_ by controlling the nucleation and growth mechanism. By introducing SDS to the electrolyte, the surfactant could be absorbed on the Mn(OH)_2_ surface under the electric field force. The interaction between the surfactant and Mn(OH)_2_ surface could be controlled in the thin interfacial region by electrochemical deposition; the Mn^2+^ ions were incorporated at the tip of the Mn(OH)_2_ coatings forming the nanowire-shaped morphology as shown in Figure 1a. It can clearly be observed that large wire-shaped nanostructures are vertically aligned on the carbon fiber, with a length more than 5 μm. The enlarged SEM image is also shown in Figure 1b in which the diameter of the nanowire is about 25 nm. In contrast to our present work, without the surfactant of SDS in the electroplating solution, the deposit is a thin film composed of nanosheets [31,32]. It can be concluded that the introduction of SDS changes the morphology and structure during the electrochemical deposition process. Compared with the nanosheets, the vertically aligned Mn(OH)_2_ nanowires expose a larger specific surface area and more active sites. This morphology and structure are helpful for the intercalation and de-intercalation of Zn^2+^. After the deposition, the surfactants can be easily removed from the surface of the nanowires with alcohol. Figure 1c shows the XRD pattern of Mn(OH)_2_ (for comparison, the XRD pattern of the CC substrate is also provided). It is obvious that a high amorphous peak centered at 25.52° appears, which comes from the CC substrate. The peaks at 2θ = 18.97° and 36.82° correspond to the (0 0 1) and (0 1 1) planes of Mn(OH)_2_ (JCPDS No: 73-1133), respectively.

The morphology and microstructure of the prepared sample were further investigated by TEM. The loose structure of SDS associated with Mn(OH)_2_ nanowires could be verified by the TEM image shown in Figure 2a. Figure 2b shows a high-resolution TEM image of Mn(OH)_2_ nanowires and the corresponding selected area electron diffraction (SAED). The SAED pattern presents a periodic structure indicating the single-crystal character of the sample. The high-resolution TEM image shows clear lattice fringes, and the measured d-spacing is 0.515 nm, corresponding to the (001) plane of Mn(OH)_2_. Figure 2c shows a typical high-angle annular dark field (HAADF) image of the loose nanowire bunch; the EDS mappings (Figure 2d,e) and spectrum (Figure 2f) were collected from the central rectangular area. Only Mn and O signals can be detected in the EDS and the elements of Mn and O distribute uniformly in the entire scanned aera, which reflects that the sample is Mn(OH)_2_ indirectly.

XPS was also used to analyze the valence states and chemical composition of the samples. The XPS spectrum shows significant peaks of Mn and O (Figure 3a) indicating that the sample is composed of Mn and O elements, which further confirms that the sample is Mn(OH)_2_ (the peak of C is from the carbon cloth substrate). Figure 3b shows the core level spectrum of the Mn 2p state. There are two sets of spin-orbital peaks at the binding energies of 653.3 and 641.9 eV, corresponding to Mn 2p^1/2^ and Mn 2p^3/2^ spin-orbital peaks [39,40,41]. The average oxidation state of Mn can also be estimated by the binding energy (ΔE_b_) between two Mn 3s peaks. The Mn 3s core-level spectrum in Figure 3c manifests that the peak splitting of the doublet is ≈5.7 eV, which is between 5.41 and 6.1 eV for Mn^3+^ and Mn^2+^, respectively [42]. The estimated average valence state of manganese in the sample is about 2.2, which indicates the partial oxidation of the sample. To evaluate the exact valence of Mn, the O 1s spectrum was also analyzed. The core-level O 1s XPS spectrum can be deconvoluted into three sub-peaks, as show in Figure 3d. Three obvious peaks with binding energies of 530.2, 531.54 and 532.2 eV are assigned to anhydrous manganese oxides (Mn-O-Mn bond), hydrated manganese oxides (Mn-O-H bond) and residual structure water (H-O-H bond), respectively [40,41]. The presence of Mn-O-H and H-O-H bonds indicates that a large quantity of crystallized water of is embedded in the sample.

The electrochemical performance of the product was investigated systematically within a typical 2032-coin cell at room temperature. Figure 4a presents the CV curves of Mn(OH)_2_ with a scan rate of 1 mv^−1^ in a potential window of 0.3–1.9 V from the first to the third cycle. One oxidation peak and two reduction peaks are obviously observed. The oxidation peak at 1.60 V resulted from the extraction of Zn^2+^ from the cathode material. For the cathodic process, the two-step charge storage was attributed to the different insertion mechanisms, the first cathodic peak located at 1.38 V was attributed to H+ intercalation, while the second cathodic peak located at 1.13 V was caused by Zn^2+^ intercalation [6]. Obviously, the first cycle was quite different from the last two, which may have been due to the activation process of the battery. However, the last two cycles basically overlapped, which indicated that the electrochemical process of the prepared sample was highly reversible. Figure 4b shows the constant current charge–discharge curves at different current densities from 0.1 to 1.0 Ag^−1^. It can be observed from the curves that there are two charge–discharge platforms; the first plateau is caused by the Zn^2+^ intercalation and the second plateau is attributed to the H^+^ intercalation [6]. The energy density of the SDS-Mn(OH)_2_ cathode was 197.6 W h kg^−1^ at a current density of 0.1 Ag^−1^. The rate performance of the product is shown in Figure 4c. Mn(OH)_2_ delivered excellent reversible rate capacities of 146.3, 128.5, 114.2, 101.4, 94.5 and 88.2 Ma h g^−1^ at the current densities of 0.1, 0.2, 0.3, 0.5, 0.7 and 1.0 Ag^−1^, respectively. The value is comparable to an MnO_2_ nanofiber [29], Mn(OH)_2_@ porous Ni [38], V_2_O_5_/CNT [21] and PANi [25]. More importantly, when the rate returns to 0.1 Ag^−1^, the specific capacity of the sample retains 94.8% of the initial capacity, indicating good layer structure stability and excellent electrochemical reversibility. Figure 4d shows the cyclic stability of SDS-Mn(OH)_2_ in ZIBs; the discharge capacity retention of the battery is 61.1% with a coulombic efficiency near 100% at 0.5 Ag^−1^ after 400 charge and discharge cycles, substantiating the impressive durability of the Mn(OH)_2_ cathode. The capacity loss may be due to the collapse of the network of nanowires during the cycle (Appendix A).

In order to understand the Zn^2+^ storage behaviors of Mn(OH)_2_ nanowire arrays in ZIBs step by step, it is necessary to further investigate the electrochemical kinetics of the sample. A series of tests were performed to quantitatively distinguish the diffusion contribution and the surface capacitance contribution of the product. Figure 5a shows the CV curves at different scan rates from 0.1 to 1 mV s^−1^ under a voltage window of 0.2–1.9 V. The measured CV curves have similar shapes. As the scan rate increases, the reduction and oxidation peaks slightly shift to the high and low potentials, respectively. For convenience, the redox peaks were marked as peak 1, peak 2 and peak 3, as shown in Figure 5a. According to the previous studies [43,44,45,46], ion electrochemical reaction kinetics can be decided by the relationship of peak current (i) and scan rate (*v*), as described below:$$i=av^{b}$$
here i and *v* represent the peak current and the scan rate, respectively. The constants a and b are empirical parameters. From the linear relationship between log(i) and log(*v*), one can estimate the value of *b*. In particular, the *b* value can offer a critical insight into the energy storage mechanism; *b* = 0.5 indicates a diffusion-controlled process, while *b* = 1 presents surface capacitive-controlled or pseudocapacitive behavior dominated reaction. As shown in Figure 5b, the *b* values of peak 1, peak 2 and peak 3 are 0.569, 0.592 and 0.459, respectively, which are closer to 0.5, indicating that the zinc-ion insertion reaction is a process dominated by diffusion restriction. We can also calculate the specific contribution of each part by the equation i = k_1_*v* + k_2_*v*^1/2^. Here, the current at a specific potential is divided into two parts: k_1_*v* represents the capacitive contribution and k_2_*v*^1/2^ represents the diffusion effect. Figure 5c shows the proportion of the capacitance contribution at the scan rate of 0.1 mv^−1^, which is calculated to be 35.1%. This reveals that the diffusion dominates the total volume at a low scan rate. In addition, by changing the scan rate from 0.1 to 1 mv^−1^, the capacitance contribution proportion increases from 35.1% to 82.3%, as shown in Figure 5d, which indicates that the capacitive effect plays a dominant role at high scan rates.

To study the resistance and the diffusivity of the sample, the Nyquist diagram and the equivalent circuit are shown in the illustration in Figure 5e. The Z’ intercept includes the equivalent series resistance R_s_ and the resistance of the electrode. The resistance of the Mn(OH)_2_ nanowire array is 6.3 Ω and 10.4 Ω before and after the cycle, indicating that Mn(OH)_2_ has a low resistance. The semicircle in the CR2032 battery diagram is related to charge transfer resistance (R_ct_). The value of R_ct_ after circulation reduces slightly, compared to that before circulation (298.4 Ω vs. 305.2 Ω), indicating that the insertion of Zn^2+^ activates the material and the electrode has good contact with the electrolyte interface. It can be concluded that the Mn(OH)_2_ nanowire array has a low charge transfer resistance, resulting in good rate and electrochemical performances. To further investigate the original charge transfer kinetics, the Nyquist diagrams collected before and after cycles are shown in Figure 5e. Two features are evident: a semicircle in the mid–high frequency region and a steep straight line in the low-frequency region. The impedance data can be fitted according to the equivalent circuit inserted in Figure 5e; the parameters R_s_, R_ct_ and Z_w_ represent the series resistance, charge-transfer resistance and Warburg diffusion process. The R_s_ value slightly increased from 6.3 to 10.4 Ω after the cycle, indicating that Mn(OH)_2_ has a low resistance. The R_ct_ value slightly decreased from 305.2 to 298.4 Ω after cycling; the results indicate an activation process in the initial several cycles, which originated from the electrolytes soaked into the electrode surface, making the redox reaction on the entire surface easier. While the slope of the straight-line part in the low-frequency region decreased after the cycles, this indicated that the ionic diffusion rate degraded within the electrode, which caused the formation of solid electrolyte interphase (SEI) growth on the electrode’s surface. In addition, GITT was employed to study the dynamics of Zn^2+^ ion diffusivity (the calculated details are shown as the Supporting Notes in the Appendix A). As shown in Figure 5f, the diffusion coefficients of Mn(OH)_2_ nanowire arrays are 4.5 × 10^−8^~1.0 × 10^−9^ cm^2^ s^−1^ and 1.0 × 10^−9^~2.7 × 10^−11^ cm^2^ s^−1^ for charging and discharging processes, respectively, reflecting the rapid ionic diffusion kinetics. Compared with other referenced cathode materials (Appendix A), the diffusion coefficients of Mn(OH)_2_ nanowire arrays were higher than the Zn^2+^ ion diffusion coefficient in MnO_2_ [16,17,18,19,47,48], V_2_O_5_ [20,21,49,50,51] and that in other related materials [24,25,52]. The high Zn^2+^ ion diffusion coefficients also benefited from the specific configuration in which the vertically aligned arrangement was beneficial to electron transport and the free space between the nanowires, which can provide more ion-diffusion pathways.

## 4. Conclusions

In conclusion, binder-free electrode Mn(OH)_2_ nanowire arrays were facilely synthesized via electrodeposition. The nanowires were vertically aligned on a carbon cloth. The as-prepared Mn(OH)_2_ nanowire arrays were used as cathodes to fabricate high-performance rechargeable aqueous ZIBs. Benefitting from their vertically aligned configuration, the electrons can be transported along the axial direction and the ions can diffuse in the free space between the nanowires. As a result, Mn(OH)_2_ nanowire arrays yielded a high specific capacitance of 146.3 mA h g^−1^ at a current density of 0.5 A g^−1^. Furthermore, the rechargeable Zn//Mn(OH)_2_ battery presented a good capacity retention of 61.1% of the initial value after 400 cycles. Our study presented an inspiring solution to develop a binder-free electrode through a simple surfactant-assistant electroplate method. This study opens a new avenue to boost the electrochemical kinetics for high-performance aqueous ZIBs.

## Figures and Tables

**Figure 1 nanomaterials-12-02514-f001:**
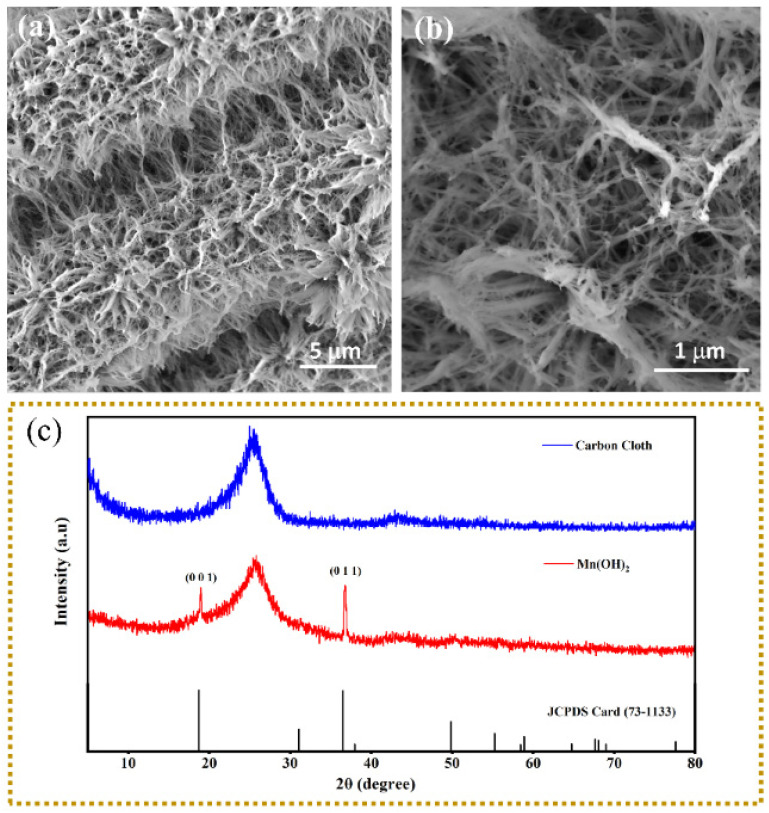
(**a**,**b**) SEM images; (**c**) XRD pattern of as-prepared Mn(OH)_2_ nanowire arrays.

**Figure 2 nanomaterials-12-02514-f002:**
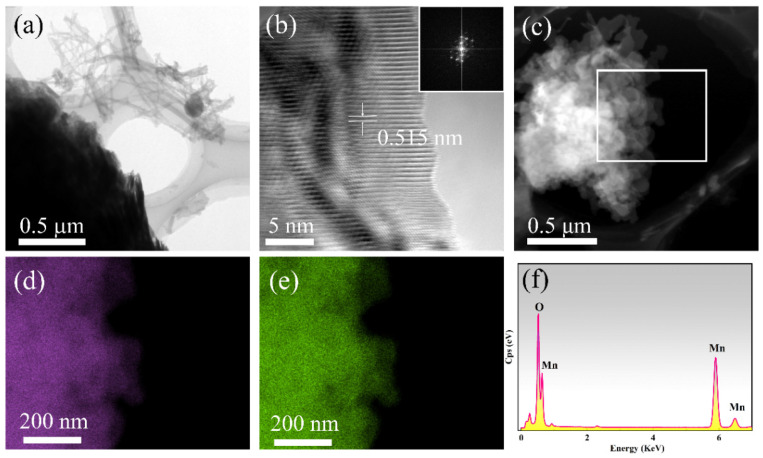
(**a**) TEM image, (**b**) HRTEM image and SAED pattern, (**c**) HAADF image of Mn(OH)_2_ nanowires. (**d**–**f**) Elemental mappings of Mn and O and EDS spectrum of the products.

**Figure 3 nanomaterials-12-02514-f003:**
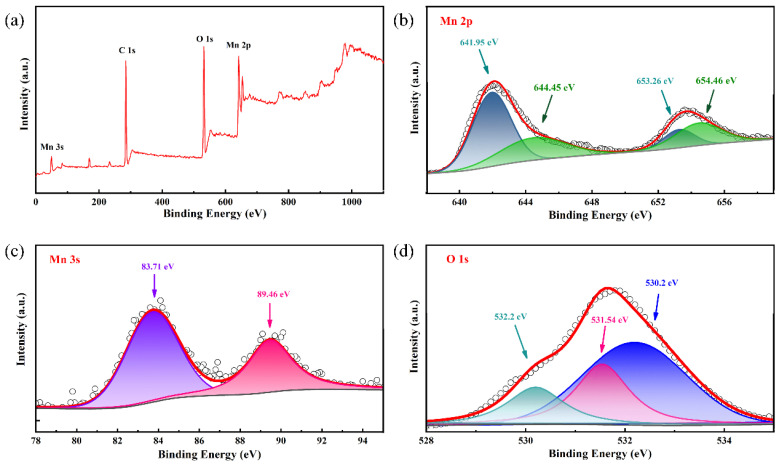
(**a**) The XPS spectra, (**b**) Mn 2p, (**c**) Mn 2s and (**d**) O 1s core-level XPS spectra of as-prepared Mn(OH)_2_ nanowire arrays.

**Figure 4 nanomaterials-12-02514-f004:**
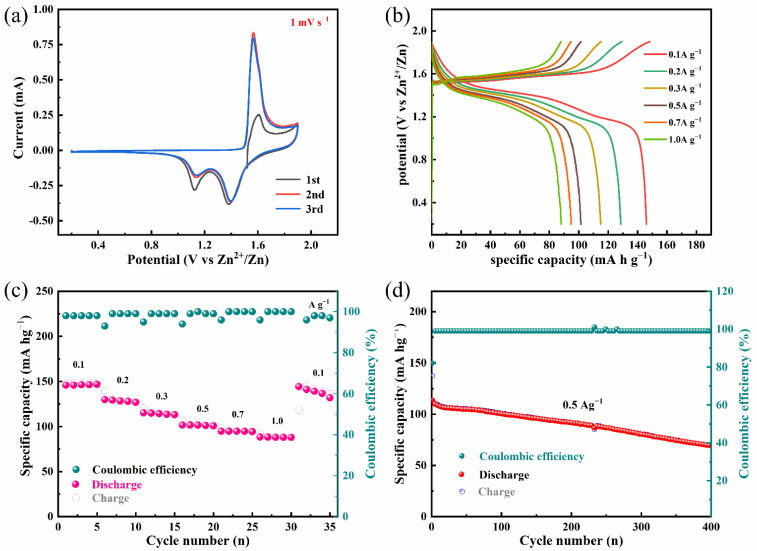
Electrochemical performance of Mn(OH)_2_//Zn ZIB in coin cells. (**a**) CV curves scanned at 1 mV s^−1^. (**b**) GCD curves tested at various current densities (0.1–1 Ag^−1^). (**c**) Rate performance detected at various specific currents from 0.1 to 1.0 Ag^−^^1^. (**d**) Cycling stability of the Mn(OH)_2_//Zn coin cells cycled at 0.5 Ag^−1^.

**Figure 5 nanomaterials-12-02514-f005:**
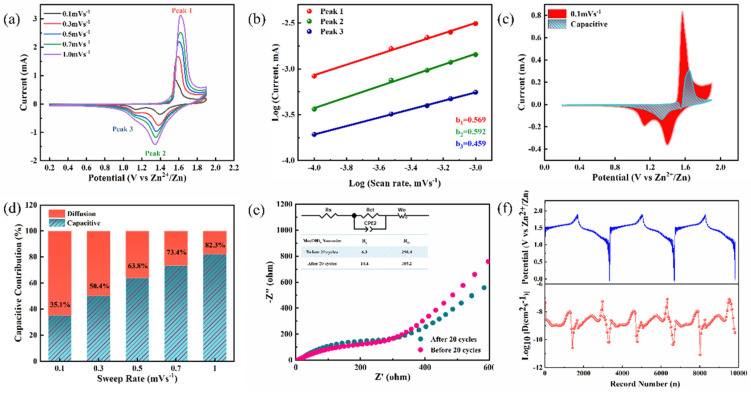
(**a**) CV curves at different scan rates. (**b**) ln(i) versus ln(v) plots at specific peak currents. (**c**,**d**) The contribution ratio of the capacitive capacities and diffusion-limited capacities under different scan rates. (**e**) Nyquist plot profiles and (**f**) charge–discharge GITT of Mn(OH)_2_//Zn ZIB.

## Data Availability

Not applicable.

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
