# Peer review of "The Synthesis of Manganese Hydroxide Nanowire Arrays for a High-Performance Zinc-Ion Battery"

_nanomaterials, 2022, doi:10.3390/nano12152514_

Round 1

Reviewer 1 Report

The manuscript reported Mn(OH)2 nanowire array decorated carbon cloth fabricated by electrodeposited for aqueous zinc ion batteries. Benefiting from the unique structure, the sample showed a high capacitance and a high diffusion coefficient. The following issues are needed to be addressed before publication.

1. The reaction of formation of Mn(OH)2 should be provided.

2. The authors claimed that “The spin-energy separation is 11.4 eV, which is in accordance with the previous reports on MnO2.”, which can’t confirm the formation of Mn(OH)2 and the fitting results were necessary to explain。

3. Does the Zn2+ and H+ storage mechanism of Mn(OH)2 has been reported? Please provide the reference or other experimental proof.

4. The morphology of the electrode after cycling was suggested to provide to check the structural stability of the electrode.

5. The related literature was suggested to cite (Rare Metal, 2022, 41, 415-424; Energy Storage Mater. 2022, 47, 98-104; ACS Appl. Mater. Interfaces 2021, 13, 14, 16869-16875.).

6. Does the carbon cloth contribute to the capacity?

7. The inset of Fig. 5e is not clear to see.

8. The electrochemical performance comparison of the sample with literature was suggested to provide to highlight the advantage of this work.

Author Response

Dear editor Johnny Wang

Thank you for your letter dated on July 8th, 2022. We have carefully read the reviewers’ comments and revised our manuscript according to their opinions. Two copies of the final manuscript file have been provided. One file that does not contain any highlighting or editing marks is uploaded as the “Primary Manuscript”. The other highlights the changes made on revision and serves as “Supporting Information for Review”. Here, we would like to answer the reviewers’ comments point by point.

Now, we upload the revised manuscript to you. Your reviewing and consideration for publishing our manuscript are greatly appreciated. If you have any questions about the revised manuscript, please don’t hesitate to contact us by e-mail: [email protected]. We would be much obliged if you could give us an early reply at your convenience.

Yours sincerely,

Prof. J.F. Gong

College of Science, Hohai University

Referee 1: Comments and Suggestions for Authors

The manuscript reported Mn(OH)2 nanowire array decorated carbon cloth fabricated by electrodeposited for aqueous zinc ion batteries. Benefiting from the unique structure, the sample showed a high capacitance and a high diffusion coefficient. The following issues are needed to be addressed before publication.

  1. The reaction of formation of Mn(OH)2should be provided.

Author Reply:

In some works, the electrodepositions were conducted in an acidic sulfate solution, which proceeds through the following reactions:

At Anode: Mn2+ + 2H2O → MnO2 + 4H+ + 2e

At Cathode: 2H+ + 2e− → H2

However, in our work, neutral nitrate solutions were used for as the electrolyte, the main reaction was described as follow. First, NO3 ions were reduced to NH4+ with generated plenty of OHin the solution. Then Mn2+ reacted with OHformed Mn(OH)2.

NO3 + 7H2O +8e→ NH4+ + 10OH

Mn2+ + 2OH→ Mn(OH)2

The deposition mechanism was also proved by Xu et.al. (Ionics 2019, 25, 3287-3298.)

  1. The authors claimed that “The spin-energy separation is 11.4 eV, which is in accordance with the previous reports on MnO2.”, which can’t confirm the formation of Mn(OH)2 and the fitting results were necessary to explain.

Author Reply:

By compared the XPS results of Mn(OH)2 and reported MnO2, we found that the spin-energy separation of Mn 2p1/2 and 2p1/2 in Mn(OH)2 is almost the same with MnO2, so we do this statement in the paper. According to the reviewer's reminder, we find that this may be misleading the reader. In the revised manuscript, it is deleted “The spin-energy separation is 11.4 eV, which is in accordance with the previous reports on MnO2.”

  1. Does the Zn2+and Hstorage mechanism of Mn(OH)2 has been reported? Please provide the reference or other experimental proof.

Author Reply: Thank you for your useful suggestions.

Yes, Zn2+ and Hstorage mechanism in manganese oxides had been demonstrated by many scientists. For example, Prof Yao Yonggang and Xia Yongyao had demonstrated a co-insertion mechanism of H+ and Zn2+ in PANI-intercalated MnO2 nanolayers with a self-regulating function in the electrolyte, the details can be found in (Huang, J.; Wang, Z.; Hou, M.; Dong, X.; Liu, Y.; Wang, Y.; Xia, Y., Polyaniline-intercalated manganese dioxide nanolayers as a high-performance cathode material for an aqueous zinc-ion battery. Nature Communications 2018, 9, 2906). Manganese hydroxide have similar structures with manganese dioxide, We believe that the Mn(OH)2 followed the similar electrochemical storage mechanism with MnO2.

  1. The morphology of the electrode after cycling was suggested to provide to check the structural stability of the electrode.

Author Reply: Many thanks for the reviewer's reminding

In the revised manuscript, the morphology of the electrode after cycling had been added as show in Figure S1. It is observed that the network of nanowires partially collapsed after 400 charge/discharge cycle, which is the reason of the capacity loss.

  1. The related literature was suggested to cite (Rare Metal, 2022, 41, 415-424; Energy Storage Mater. 2022, 47, 98-104; ACS Appl. Mater. Interfaces 2021, 13, 14, 16869-16875.).

Author Reply:  The recommended works had been cited as reference 9,10, 22.

  1. Does the carbon cloth contribute to the capacity?

Author Reply:

The carbon cloth contributes about 10 mAh cm−1, the electrochemical capacity of Mn(OH)2 based aqueous zinc-ion battery displayed in the text had subtracted the contribution of carbon cloth.

  1. The inset of Fig. 5e is not clear to see.

Author Reply: Many thanks for the reviewer's reminding

We provide the higher resolution images in figure 5e in the revised manuscript.

  1. The electrochemical performance comparison of the sample with literature was suggested to provide to highlight the advantage of this work.

Author Reply: Thank you for your useful suggestions.

In this work, we reported the synthesis of vertically-aligned Mn(OH)2 nanowires on carbon cloth. By compare the works prepared with random coating films, our work shows great advantages. First, the vertically-aligned configuration is beneficial to the electron transport. For the vertically-aligned Mn(OH)2 nanowires, electron can transport along the axial direction of the nanowire. For the coating films, it's a long way for electrons transport to the active sites during the redox reaction. The transmission path for vertically-aligned Mn(OH)2 nanowires is much shorter than that in coating films. Second, the free space between the nanowires can provide more ion diffusion pathways. there are large space between the nanowires, the electrolyte can contact closely with active materials, the Zn2+ can intercalate/de-intercalate into the Mn(OH)2 lattice along the radial direction, the intercalate depth is much smaller that random distributed nanoparticles, result in the improved Zn2+ ion diffusivity.

In fact, our work provides much fast Zn2+ storage dynamics by compared with reported works. To highlight the advantage of this work, the diffusion coefficients of Zn2+ in referenced cathode materials were show in table S1 for comparation. Furthermore, the capacity of vertically-aligned Mn(OH)2 nanowires is also compared with reported works in the revised manuscript. 

Reviewer 2 Report

This submission reports successful synthesis of nanowires for Zn-ion battery. Based on the submission diffusion coefficients are high with reasonable good cyclic performance. The contents are well suitable for this journal but some amendments are necessary.

1. Introduction seems only a papers' collection without clarification why alpha-MnO2 was widely studies and advantages and disadvantages. Refer to the paper "Mn3O4@MnS composite nanoparticles as cathode materials for aqueous rechargeable Zn ion batteries" Functional Materials Letters, Vol. 14 (2021), Article Number 2143002, DOI 10.1142/S1793604721430025 for more discussions. A mini review "Toward dendrite-free alkaline zinc-based rechargeable batteries: A minireview" of the same journal Vol 12 (2019), Article Number 1930004, DOI 10.1142/S1793604719300044 is also useful for understanding, and "MXene for aqueous zinc-based energy storage devices", Vol 14 (2021),  Article Number 2130011, DOI 10.1142/S1793604721300115

2.Use of binder-free electrode Mn(OH)2 nanowire arrays is a good idea. However, on the other hand, authors need to clarify the main cause of high diffusivity. Is it due to intrinsic diffusion path or remove of binder?

3. From SEM, it seems to me that it is not an array type of nanowire while it is a randown nanowire networks.

4. It is not clear from the main text about the redox reaction of Zn battery. Referring fig. 4 a, what are the redox reactions during charge and discharge? Why is there one anodic reaction but two cathodic ones?

5. There exists confusion about EIS. What are Rs and the resistance of the electrode? Later there is a charge transfer resistance, Rct. Is the resistance of the electrode the same as Rct and what is cause of Rs.

Author Response

Dear editor Johnny Wang

Thank you for your letter dated on July 8th, 2022. We have carefully read the reviewers’ comments and revised our manuscript according to their opinions. Two copies of the final manuscript file have been provided. One file that does not contain any highlighting or editing marks is uploaded as the “Primary Manuscript”. The other highlights the changes made on revision and serves as “Supporting Information for Review”. Here, we would like to answer the reviewers’ comments point by point.

Now, we upload the revised manuscript to you. Your reviewing and consideration for publishing our manuscript are greatly appreciated. If you have any questions about the revised manuscript, please don’t hesitate to contact us by e-mail: [email protected]. We would be much obliged if you could give us an early reply at your convenience.

Yours sincerely,

Prof. J.F. Gong

College of Science, Hohai University

Referee 2: Comments and Suggestions for Authors

This submission reports successful synthesis of nanowires for Zn-ion battery. Based on the submission diffusion coefficients are high with reasonable good cyclic performance. The contents are well suitable for this journal but some amendments are necessary.

  1. Introduction seems only a papers' collection without clarification why alpha-MnO2 was widely studies and advantages and disadvantages. Refer to the paper "Mn3O4@MnS composite nanoparticles as cathode materials for aqueous rechargeable Zn ion batteries" Functional Materials Letters, Vol. 14 (2021), Article Number 2143002, DOI 10.1142/S1793604721430025 for more discussions. A mini review "Toward dendrite-free alkaline zinc-based rechargeable batteries: A minireview" of the same journal Vol 12 (2019), Article Number 1930004, DOI 10.1142/S1793604719300044 is also useful for understanding, and "MXene for aqueous zinc-based energy storage devices", Vol 14 (2021),  Article Number 2130011, DOI 10.1142/S1793604721300115

 Author Reply: Thank you for your useful suggestions.

According to the referee`s suggestion, we introduced the advantages and disadvantages of MnO2 in the introduction section as following. The recommended works had been cited as reference 11, 12, 23.

“It is a system engineering to promote the development of ZIBs with huge commercial prospects. To improve the intrinsic safety of ZIBs, some scientists devote to suppress the growth of zinc dendrites9-11. Current research efforts are mainly focused on exploring high performance cathode materials of ZIBs, such as manganese-based material12-19, vanadium-based material20-21, and so on22-25. Manganese based oxides are the most favorable cathode materials for aqueous ZIBs because of these advantages. First, manganese ion consists with various oxidations states (Mn2+, Mn3+, Mn4+), MnO2 can accommodate one Zn2+ insertion per formula with a high theoretical capacity of approximately 616 mAh/g, in which the Mn4+is reduced to Mn2+.14 Second, MnO2 exists in various crystal forms. In these structures, the basic structural unit MnO6 octahedra is connected to each other by co-angle/co-edge, constructing chain, tunnel, layered structures with enough space accommodating foreign cations. The advantages of these structures were the high availability of Zn2+ storage site, increased electrolyte penetration and more favorable surface/interfacial properties26.”

2.Use of binder-free electrode Mn(OH)2 nanowire arrays is a good idea. However, on the other hand, authors need to clarify the main cause of high diffusivity. Is it due to intrinsic diffusion path or remove of binder?

Author Reply: Thank you for your useful suggestions.

Usually, the working electrode was fabricated by compressing a mixture of the active materials, the conductive material (acetylene black), and the binder (polytetrafluoroethylene) in a certain weight ratio. During the redox reaction, it's a long way for electrons transport to the active sites. Due to the poor conductivity of the active materials, acetylene black is used to enhanced the conductivity of coating films. However, when the active materials perpendicular to current collector, the electron can transport along the axial direction of the nanowire, which is much shorter than that in coating films. Furthermore, there are large space between the nanowires, the electrolyte can contact closely with active materials, the Zn2+ can intercalate/de-intercalate into the Mn(OH)2 lattice along the radial direction, the intercalate depth is much smaller that random distributed nanoparticles, result in the improved Zn2+ ion diffusivity. The schematic diagram is displayed as below.

  1. From SEM, it seems to me that it is not an array type of nanowire while it is a randown nanowire networks.

Author Reply:

I agree with your viewpoint. It seems a disordered nanowire network form SEM image. The reason for this is that the surface tension of liquid in the process of liquid volatilization, which cause the interspace between the nanowires collapses at solid-liquid interface. We can assemble the cells without drying the Mn(OH)2/CC substrate. Under this circumstances, most of the Mn(OH)2 nanowire should aligned vertically on the CC.

  1. It is not clear from the main text about the redox reaction of Zn battery. Referring fig. 4 a, what are the redox reactions during charge and discharge? Why is there one anodic reaction but two cathodic ones?

Author Reply:

In figure 4a, the CV curves show one anodic peak and two cathodic peaks. The asymmetric redox reaction is common phenomenon in manganese oxide. It is proved that the oxidation peak at 1.60 V resulted from the extraction of Zn2+ from cathode material. For the cathodic process, the two-step charge storage is attributed to the different insertion mechanism, the first cathodic peak is caused by Zn2+ intercalation and the second cathodic peak is attributed to H+ intercalation (Nature Communications 2018, 9, 2906).

  1. There exists confusion about EIS. What are Rs and the resistance of the electrode? Later there is a charge transfer resistance, Rct. Is the resistance of the electrode the same as Rct and what is cause of Rs.

 Author Reply: Thank you for your useful suggestions.

We rewrite the EIS section, the Rs, Rct and the reasons why they changed were clarified in the revised manuscript as following: “To further investigate the original charge transfer kinetics, Nyquist diagram collected before and after cycles are shown in figure 5e. Two features are evident: a semicircle in the mid-high frequency region and a steep straight line in the low frequency region. The impedance data can be fitted according to the equivalent circuit insert in figure 5e, the parameters Rs, Rct and Zw represent to the series resistance, charge-transfer resistance and Warburg diffusion process. The Rs value increased slightly from 6.3 to 10.4 Ω after the cycle, indicating that Mn(OH)2 has a low resistance. The Rct value decreased slightly from 305.2 to 298.4 Ω after cycling, the results indicate an activation process in the initial several cycles, which originated from the electrolytes soaked into the electrode surface, making the redox reaction on the entire surface will become easier. While the slope of the straight-line part in low-frequency region decreased after the cycles, indicate ionic diffusion rate degrades within the electrode, which attribute the formation solid electrolyte interphase (SEI) growth at the electrode surface.”

Round 2

Reviewer 1 Report

Well revised manuscript. I would like to recommend its acceptance now.